Knockdown of CDCA8 inhibits the proliferation and enhances the apoptosis of bladder cancer cells

Gao Xin 1 2
Wen Xiaohong 1
He Haowei 1
Zheng Linlin 1
Yang Yibo 2
Yang Jinlian 2
Liu Haifang 1
Zhou Xiguo 2
Yang Changshun 2
Chen Yinyi 1 3
Chen Mei 1
Zhang Shufang 1 haikouyiyuan@126.com
1 Central Laboratory, Affiliated Haikou Hospital of Xiangya Medical College, Central South University , Haikou, Hainan , China
2 Clinical Laboratory, The First People’s Hospital of Huaihua of University of South China , Huaihua , China
3 Clinical Laboratory, The First Affiliated Hospital of University of South China , Hengyang , China
Della Corte Carminia
Electronic publication date: 2020 Apr 28
Publication date: 2020
Volume: 8
Electronic Location ID: e9078
Received 2019 Nov 19; Accepted 2020 Apr 7
Copyright: © 2020 Gao et al.
Copyright year: 2020
Copyright holder: Gao et al.
License: This is an open access article distributed under the terms of the Creative Commons Attribution License, which permits unrestricted use, distribution, reproduction and adaptation in any medium and for any purpose provided that it is properly attributed. For attribution, the original author(s), title, publication source (PeerJ) and either DOI or URL of the article must be cited.
License URL: https://creativecommons.org/licenses/by/4.0/

Keywords: Cell division cycle associated 8, Bladder cancer, Prognosis, Proliferation, Therapeutic target

Funding: Hainan Provincial Natural Science Foundation of China 819QN387, 2017CXTD010 and 20168312 National Science Foundation of China 81760465 Finance Science And Technology Project Of Hainan Province ZDYF2019163 and ZDKJ2017007 This study was supported by the Hainan Provincial Natural Science Foundation of China (Grant No. 819QN387, 2017CXTD010, 20168312), the National Science Foundation of China (Grant No. 81760465), and the Finance science and technology project of hainan province (Grant No. ZDYF2019163 and ZDKJ2017007). The funders had no role in study design, data collection and analysis, decision to publish, or preparation of the manuscript.

==============================
Bladder cancer is a tumour of the urinary system with high mortality, and there is also a great lack of therapeutic targets in the clinic. Cell division cycle associated 8 (CDCA8), an important component of the vertebrate chromosomal passenger complex, is highly expressed in various tumours and promotes tumour development. However, the role of CDCA8 in bladder cancer is not fully understood. This study aimed to reveal the function of CDCA8 in bladder cancer by determining the relationship between CDCA8 expression and proliferation, metastasis and apoptosis of bladder cancer cells. Firstly, we studied the mRNA expression of CDCA8 through the Gene Expression Omnibus (GEO) and the Cancer Genome Atlas (TCGA) databases and analysed the correlation between CDCA8 expression and prognosis of patients with bladder cancer. We also verified CDCA8 expression in bladder cancer tissues by immunohistochemistry. In addition, CDCA8 expression was inhibited in bladder cancer T24 and 5637 cells, and the effects of CDCA8 on the proliferation, migration and invasion of bladder cancer cell lines were investigated using cell counting kit-8, colony formation, cell cycle, apoptosis, wound healing and Transwell invasion assays. Results showed that CDCA8 was highly expressed in bladder cancer compared with normal tissues, and the high CDCA8 expression was significantly correlated with the poor prognosis of patients. Inhibiting CDCA8 expression inhibited the proliferation, migration and invasion of T24 and 5637 cells and induced the apoptosis of bladder cancer cells. CDCA8 was involved in the regulation of the growth cycle of bladder cancer cells. Bioinformatics-based mechanism analysis revealed that high CDCA8 expression may affect the cell cycle and P53 signalling pathways. In conclusion, our results suggest that CDCA8 is highly expressed in bladder cancer and can promote tumour development. Hence, CDCA8 may serve as an effective therapeutic target for treatment of bladder cancer.

Introduction

Bladder cancer is a common malignant tumour in the urinary system and has high mortality and recurrence rates. The incidence of bladder cancer in women and men ranks in the top 20 and 10 among all malignant tumours, respectively (Burger et al., 2013). Nonmuscle invasive tumours account for approximately 70% to 80% of primary bladder tumours (Alfred et al., 2017). Although first diagnosed bladder cancer is noninvasive early tumours, 1/3 of nonmuscle invasive bladder cancer metastasises to other organs and eventually develops into muscle invasive bladder cancer, resulting in poor prognosis (Van Rhijn et al., 2009). The transurethral resection of bladder tumour is a common surgical method for noninvasive bladder cancer; however, the recurrence rate is high after the operation. The recurrence rate of patients with high-grade bladder cancer can reach 50% within 1 year after operation (Lightfoot et al., 2014). At present, cystoscopy is a gold standard for diagnosis of bladder cancer, but it is complicated, costly and invasive (Geavlete et al., 2012). Urine cytology is also one of the first choices for diagnosis of bladder cancer. However, this method is less sensitive to the diagnosis of low-grade bladder cancer (Yafi et al., 2015). These conditions highlight the urgent need for new biomarkers to accurately predict the recurrence of bladder cancer. Various tumour biomarkers have been introduced into daily clinical practices, including risk assessment, differential diagnosis, prediction of treatment response and monitoring of disease progression (Wei, 2015). Therefore, biomarkers related to the occurrence, development and prognosis of bladder cancer must be identified.

The occurrence and development of bladder cancer are related to many factors, particularly the accumulation of genetic changes. The cell division cycle related protein family Cell division cycle associated 1–8 (CDCA1–8) is an important regulator of cell proliferation; they are involved in the growth regulation of normal and tumour cells. At present, studies on the CDCA protein family in bladder cancer are rare; however, with the deepening of research, the role of this family in bladder cancer will become increasingly clear. Harao et al. (2008) found that CDCA1 is highly expressed in various tumours, including bladder cancer. CDCA1-derived peptides can induce the proliferation of peptide-responsive cytotoxic T lymphocytes, thus killing tumour cell lines that endogenously express HLA-A2 and CDCA1. CDCA1 may be a molecular target for the diagnosis and immunotherapy of bladder cancer. Sugawara et al. (2018) found two important miRNAs, namely, mir-223 double stranded body (mir-223)-5p and mir-223-3p, by sequencing and analysing the RNA of bladder cancer tissues. The results of cell function experiments showed that mir-223 double-stranded body plays an important role in the migration and invasion of bladder cancer cells. To identify the target genes of miR-223 double-stranded body, they further mined the Cancer Genome Atlas (TCGA) database and found 20 target genes, including CDCA2. The study also found that CDCA2 is highly expressed in bladder cancer and is associated with poor prognosis. Therefore, miR-223 double-stranded body may promote the development of bladder cancer by regulating the expression of CDCA2. In addition, CDCA5 overexpression in bladder cancer promotes cancer progression, leading to poor patient prognosis; CDCA5 also participates in the G1–S cell cycle transition of bladder cancer cells (Chang et al., 2015).

The CDCA protein may be involved in the regulation of mitosis, crossover chromosome separation and division in cancer cells (Wang et al., 2014). Human cell division cycle related 8 (CDCA8) protein is an important component of vertebrate chromosome passenger complex (CPC) (Phan et al., 2018). CPC contains at least four proteins, AuroraB, INCENP, survivin and CDCA8, which play a basic regulatory role and participate in cell dynamic localisation during each mitosis (Gassmann et al., 2004; Higuchi & Uhlmann, 2003). Therefore, CDCA8 is an essential regulator of cell mitosis. Dai et al. (2015) found that CDCA8 overexpression is essential for the growth of embryonic stem cells and breast cancer cells. CDCA8 is highly expressed in most types of tumours but lowly expressed in normal tissues (Narayan et al., 2007). Studies have shown that the high CDCA8 expression is significantly associated with poor prognosis in patients with breast cancer (Jiao et al., 2015), gastric cancer (Chang et al., 2006) and kidney cancer (Gu et al., 2017). CDCA8 is also involved in the regulation of tumour cell growth. Yu et al. (2019) found that high CDCA8 expression in tamoxifen-resistant breast cancer cells promotes the growth of tamoxifen-resistant breast cancer cells and induces breast cancer cell resistance. Bu et al. (2019) investigated the role of CDCA8 in the growth of Estradiol (E2)—stimulated breast cancer cells. CDCA8 is a key mediator of E2-stimulated breast cancer cell growth and can be used as a new target in breast cancer treatment. KIF18B is overexpressed in pancreatic ductal adenocarcinoma and can promote tumour development. Importantly, KIF18B can bind to the promoter region of CDCA8, resulting in CDCA8 overexpression (Li et al., 2020). Therefore, CDCA8 plays an important role in the occurrence and development of tumour, and is expected to become a molecular target for tumour therapy and a marker for monitoring patient prognosis. However, the role of the CDCA8 in bladder cancer is unclear.

In this study, GSE13507, GSE7476, GSE37815, GSE65635 and GSE48075 datasets were obtained from the Gene Expression Omnibus (GEO) database, from which CDCA8 expression was analysed. Kaplan–Meier and COX regression method were used to analyse the correlation between CDCA8 expression in bladder cancer and prognosis. We also analysed the correlation of CDCA8 expression and prognosis in bladder cancer in the TCGA database. We verified the expression and clinical relevance of CDCA8 in bladder cancer tissues by immunohistochemistry (IHC). To further understand the role of CDCA8 in bladder cancer, we studied the functions of CDCA8 in bladder cancer cells in vitro and analysed its possible mechanisms.

Materials and Methods

Expression profile data of bladder cancer

The processed expression datasets of bladder cancer were obtained from the GEO database (https://www.ncbi.nlm.nih.gov/geo/). The details of the four bladder cancer datasets are shown in Table 1, and the clinical information of GSE13507 dataset is shown in Table 2. The above sample data are obtained from the public database and hence do not require approval by the ethics committee.

Table 1 Details of four bladder cancer sequencing datasets in the GEO database.

Publisher	ID	Platform	Normal	Tumor	
Mengual et al. (2009)	GSE7476	GPL570	3	9	
Kim et al. (2010) and Lee et al. (2010)	GSE13507	GPL6102	9	165	
Kim et al. (2013)	GSE37815	GPL6102	5	18	
Borisov et al. (2018)	GSE65635	GPL14951	4	8	
Note:

GEO, Gene Expression Omnibus.

Table 2 Clinical information of GSE13507 dataset in GEO database.

Characteristics	Patients (N = 165)	
	n	%	
Age category				
<60 years	46	27.88	
≥60 years	119	72.12	
Gender				
Male	135	81.82	
Female	30	18.18	
Progression				
No	82	49.70	
Yes	83	50.30	
Invasiveness				
Superficial	103	62.42	
Invasive	62	37.58	
Grade				
Low	105	63.64	
High	60	36.36	
T stage				
Ta	24	14.55	
T1	80	48.48	
T2	31	18.79	
T3	19	11.52	
T4	11	6.67	
N stage				
N0	150	90.91	
N1	8	4.84	
N2	5	3.03	
M stage				
M0	158	95.76	
M1	7	4.24	
Overall survival				
Alive	96	58.18	
Dead	69	41.82	
Cancer-specific survival				
Alive	133	80.61	
Dead	32	19.39	
Note:

GEO, Gene Expression Omnibus.

After obtaining the GSE13507, GSE7476, GSE37815 and GSE65635 datasets, we used Perl software (version: 5.28) to transform the gene probe ID of the dataset expression profile into the gene symbol of the platform file and obtained four gene expression profile files containing standard gene symbol. We then extracted the expression value of CDCA8 in all samples from each dataset for differential analysis. Univariate and multifactorial COX regression analysis was performed by survival package in R language. In addition, we downloaded the bladder cancer gene expression data and overall survival data from TCGA database (https://portal.gdc.cancer.gov/) and extracted the expression data and prognosis data of CDCA8 in 19 normal samples and 414 bladder cancer samples for analysis.

Immunohistochemical analysis

We collected seven normal bladder tissues and 35 bladder cancer tissues from Haikou Municipal People’s Hospital from October 2018 to December 2019. All sample collections were approved by the Haikou Municipal People’s Hospital Ethics Committee under approval number SC20180238. We placed the tissue slices into a 63 °C oven for 1 h and performed dewaxing process involving 15 min in xylene and 7 min in anhydrous ethanol. We then used the Dako automatic immunohistochemistry pretreatment system to repair antigens. After repairing the antigen, the tissues were rinsed with PBST buffer 3 times for 1 min each. In accordance with the instructions, we diluted the first antibody with the antibody diluent. After adding an antibody, the samples were incubated at room temperature for 30 min. The slices were rinsed with PBST buffer 3 times for 1 min each and then added with the second antibody. The slice was incubated with diluted DAB for 10 min and then washed for 5 min. The slice was added with haematoxylin (SIGMA) for 1 min, immersed in 0.25% hydrochloric acid alcohol (400 mL of 70% alcohol + 1 mL of concentrated hydrochloric acid) for not less than 2 s, rinsed with tap water for more than 2 min, dried at room temperature and sealed. Finally, the dyeing results were read by a pathologist. The criteria for judging dyeing intensity and percentage were as follows: intensity: negative, score 0; weak, score 1; moderate, score 2; strong, score 3; percentage: 0%, score 0; 1–10%, score 1; 11–50%, score 2; 51–100%, score 3. The two scores were multiplied to obtain the final score (Hashimoto et al., 2020; She et al., 2019).

Cell culture

Human T24 and 5637 bladder cancer cell lines were purchased from the Shanghai Institutes for Biological Sciences. These cells were cultured at 37 °C in a 5% CO2 atmosphere with DMEM containing 10% 10–15 mL of foetal bovine serum. The medium was changed every 3 days. The cells were washed 3 times with PBS and digested with 0.25% trypsin while the cells were in the logarithmic growth phase. The suspension of cells was separately inoculated into 6-well plates for the preparation for subsequent experiments.

Cell transfection

The lentivirus was transfected when the cell density in the 6-well plate reached approximately 15–30%. The lentiviral vector was synthesised by Shanghai Jikai Company. The target sequence of CDCA8 gene selected in this experiment was (si-CDCA8): 5′-TTGACTCAAGGGTCTTCAA-3′ with a GC content of 42.11%. The RNAi negative control sequence was (si-NC): 5′-TCTCCGAACGTTTCACGT-3′ with a GC content of 52.63%. The cells were transfected with MOI = 10, the amount of virus in the si-CDCA8 group and the si-NC group was 2 μL, and the virus titre was 5 × 108 TU/mL. After the transfection was completed, the cells were further cultured in a 37 °C and 5% CO2 incubator for approximately 72 h. The expression of the green fluorescent protein (GFP) in the cells was observed under a fluorescence microscope. GFP expression is used to judge transfection efficiency. In addition, we used quantitative real-time polymerase chain reaction (qRT-PCR) and Western blot (WB) to verify the effect of siRNA transfection. The transfected cells were then prepared for subsequent steps.

qRT-PCR assay

After 72 h of transfection, total RNA was extracted using 1 mL of Trizol. cDNA reverse transcription was then performed using an M-MLV kit (Promega, Shanghai, China). Approximately 2.0 μg of total RNA and 1 μL of Oligo dT (0.5 μg/μL) were added to the PCR system and then supplemented with RNase-free H2O to 10 μL. The primers were: CDCA8 forward, 5′-GCAGGAGAGCGGATTTACAAC-3′ and reverse, 5′-CTGGGCAATACTGTGCCTCTG-3′; and GAPDH forward, 5′-TGACTTCAACAGCGACACCCA-3′ and reverse, 5′-CACCCTGTTGCTGTAGCCAAA-3′. The cycle condition was 95 °C for 5 min, followed by 50 cycles at 95 °C for 30 s and 60 °C for 45 s. The last step was 72 °C for 30 min. ∆Ct = Ct value of target gene is the Ct value of internal reference gene; −∆∆Ct = mean value of ∆Ct of NC group is the ∆Ct value of each sample; the expression level of target gene in each sample relative to NC group is represented by 2−∆∆Ct (Ci et al., 2019; Livak & Schmittgen, 2001).

WB assay

After 72 h of transfection, the cells were washed with PBS 3 times, cleaved with radio immunoprecipitation assay lysis buffer, and then broken by 200 W ultrasound on the ice. The procedure was repeated 4 times, 5 s per time with an interval of 2 s. The supernatant was centrifuged for 15 min at 4 °C and 12,000×g. Protein concentration was determined by BCA reagent. Total protein was separated by SDS-PAGE, which consisted of 10% polyacrylamide sequencing gel, and then transferred to the PVDF membrane sealed with 5% skimmed milk powder TBST sealing solution for 1 h. CDCA8 antibody (Abeam, Shanghai, China) was diluted with PBS, incubated with the blocked PVDF membrane for 2 h or at 4 °C overnight at room temperature, and washed 4 times with TBST for 8 min/time. The secondary antibody was diluted with PBS, incubated with PVDF membrane for 1.5 h at room temperature, and then washed 4 times with TBST for 8 min/time. Finally, the PVDF membrane was placed in a cassette and added with A and B mixed solutions.

Cell proliferation assay

After 72 h of transfection, the cells were digested and inoculated into 96-well plates at a density of 4,000 cells/well. Cell counting kit-8 (CCK-8) (Sigma, Shanghai, China) was used to detect cell viability. In accordance with the operating instructions, the cell viability of T24 and 5637 was detected every 24 h (24, 48, 72, 96 and 120 h). Approximately 10 μL of CCK-8 reagent was added to the wells 2 h before the end of culture without changing the solution. The CCK-8 reagent was then added to the enzyme detector for 4 h to detect the OD value of 450 nm. A growth curve was then constructed. Each sample was tested 3 times.

Cell clone formation assay

The infected cells were digested with trypsin, resuspended and then counted in the complete medium. According to the number of 500 cells/holes, the cells were re-inoculated into the 6-well plate culture plate. Three multiple holes were set up in each group, the culture medium was changed every 2–3 days, and the cell state was observed. After 10 days of culture, cell growth was observed under a fluorescence microscope before the end of the experiment. The cells were then washed once with PBS, fixed with 1 ml of 4% paraformaldehyde for 30–60 min and then washed with PBS. Approximately 1,000 μL of clean crystal purple dye solution was added to each hole and allowed to stain the cells for 10–20 min. The cells were washed with ddH2O several times, dried and photographed with a digital camera. Three repeated experiments were set up in each group.

Wound healing assay

The cells were digested with trypsin, suspended in complete medium and then counted. The cell density of the 96-well plate was adjusted to 50,000 cells/well, and the volume was 100 μL/well. Three multiple wells were set up in each group, and culture medium containing 1% serum was replaced the next day. The lower centre of the 96-well plate was aligned with the scraping device and gently pushed upward to form a scratch. The cells were washed 2–3 times with serum-free complete medium, cultured with medium containing 0.5% FBS and photographed at 0 h. Cells were cultured in a 37 °C incubator with 5% CO2 and scanned with Celigo at 0, 8 and 24 h after scraping. Migration area was analysed by Celigo: migration area = Xh cell area − 0 h cell area. Three repeated experiments were set up for each sample.

Transwell assay

The chamber was placed in a new 24-well plate. Approximately 100 μL of serum-free medium was added to the upper chamber, and the plate was placed in a 37 °C incubator for 1 h. The cells in logarithmic growth phase were digested and prepared into a cell suspension with serum-free medium. The cells were counted and inoculated in a 24-well plate. The number of inoculated cells was controlled at 105/well. The medium in the upper chamber was carefully replaced with 100 μL of the cell suspension. Approximately 600 μL of 30% FBS medium was then added to the lower chamber. The cells were cultured in a 37 °C incubator for 40 h. The plate was then inverted onto an absorbent paper to remove the culture medium. The nontransferred cells in the chamber were slowly removed with cotton swabs, and the cells were fixed in 4% paraformaldehyde solution for 0.5 h. After fixation, the fixative on the surface of the chamber was absorbed with paper. The cells were then stained with 1–2 drops of staining solution under the membrane so that the transferred cells could be stained for 1–3 min. The chamber was then soaked and rinsed several times and allowed to dry naturally. Four 100× photos and nine 200× photos were randomly captured from the Transwell’s visual field. The number of metastatic cells per field was counted using 200× photos. Three repeats was performed in each group.

Cell cycle analysis

After 72 h of transfection, the cells in the logarithmic growth phase were collected, and the supernatant was discarded after centrifugation. The cells were washed with D-Hanks (pH = 7.2–7.4). After centrifugation, the cells were fixed with 75% ethanol precooled at 4 °C for at least 1 h. The cells were suspended with 0.6–1 mL of cell staining solution. The formulation of the staining solution was as follows: 40 × PI mother solution (2 mg/mL): 100 × RNase mother solution (10 mg/mL): 1 × D-Hanks = 25:10:1,000. Cell cycle was detected by flow cytometry, and the results were analysed using the ModFit software. Three repeats were applied in each group.

Apoptosis assay

At logarithmic growth stage, cells were digested and washed with D-Hanks (pH = 7.2–7.4) at 4 °C after centrifugation. The cells were washed again with 1 × binding buffer. After centrifugation, 200 μL of 1 × binding buffer was added to obtain a cell suspension, and then 10 μL of Annexin V-APC was added. The reaction was carried out in an opaque environment at room temperature for 10–15 min. Flow cytometry was used to detect apoptosis. Results were analysed using the guava InCyte software. Each group was repeated 3 times.

Statistical analysis

Statistical analysis was carried out using SPSS 23.0 software, and the counting data were tested for normality and homogeneity of variance. Normality analysis was carried out by Kolmogorov–Smirnov test. Independent sample t-test was used to compare differences between the two groups. For the counting data which did not meet the t-test conditions of independent samples, the Mann–Whitney U nonparametric test was used to compare the differences between the two groups, and the Kruskal–Wallis test was used to compare data from more than three groups. Prognostic analysis was performed using the Kaplan–Meier and Log-rank tests. The difference was statistically significant at P < 0.05.

Results

CDCA8 is upregulated in bladder cancer tissues

We extracted the expression values of CDCA8 from normal tissues and bladder cancer tissues in each dataset. The difference of CDCA8 expression between the two groups is shown in Fig. 1. Compared with normal bladder tissues, CDCA8 expression in bladder cancer tissues in the GSE7476 dataset was significantly higher than that in normal bladder tissues (Fig. 1A; P < 0.01). CDCA8 expression in bladder cancer was significantly increased in the GSE13507 dataset (Fig. 1B; P < 0.001) and in the GSE37815 dataset (Fig. 1C; P < 0.01). The results of GSE65635 dataset analysis also showed that CDCA8 expression in bladder cancer was significantly higher than that in normal tissues (Fig. 1D; P < 0.01). In the TCGA database, we obtained the same results. CDCA8 expression in bladder cancer was significantly higher than that in normal tissues (Fig. 1E; P < 0.001).

Figure 1 Analysis of CDCA8 expression and prognosis in bladder cancer.

CDCA8 expression analysis in (A) GSE7476; (B) GSE13507; (C) GSE37815 and (D) GSE65635 datasets. (E) CDCA8 expression analysis in TCGA database. (F) Analysis of correlation between CDCA8 expression in GSE13507 dataset and cancer-specific survival. (G) Analysis of correlation between CDCA8 expression in GSE13507 dataset and overall survival. (H) Analysis of the correlation between CDCA8 and the prognosis of patients with bladder cancer in TCGA database. BLCA, Bladder urothelial carcinoma. **P < 0.01, ***P < 0.001.

We analysed the correlation between CDCA8 expression and the prognosis of patients with bladder cancer in the GSE13507 dataset. Depending on the median expression of CDCA8 in bladder cancer tissues, the patients were divided into high- and low-expression patients. Cancer-specific survival analysis and overall survival analysis were carried out. The correlation between cancer-specific survival rate and CDCA8 expression is shown in Fig. 1F. The prognosis of patients with high CDCA8 expression was poor (P < 0.00028). The correlation between overall survival rate and CDCA8 expression showed the same results, and the prognosis of patients with high CDCA8 expression was poor (Fig. 1G; P < 0.0006). However, in the TCGA database, no correlation was observed between CDCA8 expression and the prognosis of patients with bladder cancer (Fig. 1H; P = 0.6).

Correlation between CDCA8 expression and the clinical characteristics of patients with bladder cancer

To further explore the effect of CDCA8 expression on the progression of bladder cancer, we obtained the CDCA8 expression value of each sample from the GSE13507 dataset and divided the samples into groups according to the clinical information of GSE13507. The difference of CDCA8 expression between groups in the same category was compared. As shown in Fig. 2, CDCA8 expression is higher in individuals with high stage and grade tumour compared with individuals with low stage and grade tumour (P < 0.05). We also found that CDCA8 expression was higher in advanced bladder cancer than in non-advanced bladder cancer (P < 0.05).

Figure 2 CDCA8 expression in bladder cancer tissues and comparison of expression among various patients.

(A) Comparison between high and low grade bladder cancer groups (P < 0.0001); (B) comparison between progression bladder cancer group and nonprogression bladder cancer group (P < 0.0001); (C) comparison between male and female patients (P > 0.05); (D) comparison between the young patient group and the elderly patient group (P = 0. 0171); (E) comparison between invasive and superficial bladder cancer groups (P < 0.0001). (F) Comparison between recurrence group and initial group (P = 0.468); (G) CDCA8 expression difference analysis between bladder cancer T stage (P < 0.0001); (H) CDCA8 expression difference analysis between bladder cancer N stage (P = 0.324); (I) CDCA8 expression difference analysis between bladder cancer M stage (P > 0.05). NS, no significance.

To understand whether the correlation between CDCA8 expression in GSE13507 dataset and the prognosis of patients with bladder cancer is independent of other clinical factors, we performed univariate COX regression and multivariate COX regression analysis on CDCA8 expression, gender, age, TNM stage and tumour grade and progression. In the results of multivariate COX regression analysis based on overall survival, no significant correlation was found between CDCA8 expression and the prognosis of patients with bladder cancer (Table 3; P = 0.114). In the results of multivariate COX regression analysis based on cancer-specific survival, CDCA8 expression was significantly associated with the prognosis of patients with bladder cancer (Table 4; P = 0.036).

Table 3 Univariate and multivariate COX regression analysis based on overall survival.

Variables	Univariate analysis	Multivariate analysis	
	HR	95% CI of HR	P-value	HR	95% CI of HR	P-value	
Age (≤79 vs. >79)	2.049	[1.066–3.939]	0.031	2.146	[1.055–4.365]	0.035	
T (Ta–T1/T2–T4)	2.73	[1.688–4.415]	<0.001	1.277	[0.678–2.407]	0.449	
N (N0/N1–N3)	8.508	[4.271–16.951]	<0.001	2.548	[0.878–7.39]	0.085	
M (M0/M1)	9.897	[4.378–22.374]	<0.001	5.984	[1.668–21.476]	0.006	
Grade (low/high)	2.74	[1.694–4.433]	<0.001	1.213	[0.656–2.242]	0.539	
CDCA8 (low/high)	2.311	[1.414–3.779]	<0.001	1.575	[0.897–2.767]	0.114	
Progression (no/yes)	2.892	[1.756–4.761]	<0.001	2.754	[1.561–4.859]	<0.001	
Gender (male/female)	0.641	[0.361–1.138]	0.129	/	/	/	

Table 4 Univariate and multivariate COX regression analysis based on tumour-specific survival.

Variables	Univariate analysis	Multivariate analysis	
	HR	95% CI of HR	P-value	HR	95% CI of HR	P-value	
Age (≤79 vs. >79)	0.912	[0.277–3.004]	0.88	/	/	/	
T (Ta–T1/T2–T4)	17.897	[6.251–51.244]	<0.001	6.569	[1.919–22.486]	0.003	
N (N0/N1–N3)	16.434	[7.557–35.738]	<0.001	3.817	[1.335–10.914]	0.012	
M (M0/M1)	13.118	[5.283–2.596]	<0.001	5.013	[1.183–21.242]	0.029	
Grade (low/high)	5.985	[2.757–12.992]	<0.001	1.002	[0.384–2.616]	0.996	
CDCA8 (low/high)	3.981	[1.781–8.9]	<0.001	2.548	[1.062–6.114]	0.036	
Progression (no/yes)	6.414	[3.184–12.922]	<0.001	5.639	[2.236–14.226]	<0.001	
Gender (male/female)	0.477	[0.22–1.032]	0.06	/	/	/	

Verification of CDCA8 expression in samples by IHC

We verified CDCA8 expression in seven normal bladder tissues and 35 bladder cancer tissues by IHC. As shown in Fig. 3, the expression score of CDCA8 in tumour tissues was significantly higher than that in normal tissues. In addition, we analysed CDCA8 expression in different clinical characteristics (gender, age, grade, lymph node invasion and T stage). The analysis results are shown in Table 5. CDCA8 expression in patients with high grade was significantly higher than that in patients with low grade (P = 0.023). CDCA8 expression in patients with high T stage was significantly higher than that in patients with low T stage (P = 0.031).

Figure 3 Immunohistochemical detection of CDCA8 expression in normal bladder tissue and bladder cancer.

(A–F) Representative photos of immunohistochemical staining. (G) IHC score of normal bladder tissue was compared with that of bladder cancer tissue. **P < 0.01.

Table 5 Analysis of the relationship between CDCA8 expression and clinical factors in patients with bladder cancer.

Parameters	n	CDCA8 protein	
		Expression	P-value	
Gender				
Male	29	7.345 ± 1.492	0.617	
Female	6	7.000 ± 1.414		
Age				
<75	17	6.882 ± 1.367	0.125	
≥75	18	7.667 ± 1.491		
Grade				
II	12	6.500 ± 1.118	0.023	
III	23	7.696 ± 1.487		
Lymph node invasion				
No	25	7.080 ± 1.440	0.206	
Yes	10	7.800 ± 1.470		
T stage				
T1–T2	19	6.789 ± 1.321	0.031	
T3–T4	16	7.875 ± 1.452		

CDCA8 knockdown in T24 and 5637 cells

We transfected T24 and 5637 cells (si-CDCA8) with lentiviruses carrying siRNA vectors and used nontargeted siRNA sequences as controls (si-NC). The total RNA of the cells was extracted after transfection, and CDCA8 mRNA expression was detected by qRT-PCR. We also analysed CDCA8 protein expression by WB. Results showed that the expression rate of GFP was more than 80% (Figs. 4A–4D), indicating that the lentivirus was successfully transfected into the cells. CDCA8 expression was significantly reduced in T24 and 5637 cells, and the knock-down efficiency of mRNA was more than 70% (Figs. 4E–4G).

Figure 4 CDCA8 expression in bladder cancer cell lines.

(A–D) GFP expression in T24 and 5637 cells under fluorescence microscope (100×). (E) T24 and 5637 mRNA expression levels of CDCA8 after transfection with lentivirus. (F and G) CDCA8 protein expression in T24 and 5637 cells transfected with lentivirus as detected by Western blot. **P < 0.01.

CDCA8 knockdown inhibits the proliferation of bladder cancer cells

After knocking down CDCA8 expression in T24 and 5637 cells, the absorptivity of wavelength at 450 nm in si-CDCA8 group and si-NC group was detected for 5 days. The value of OD450 could indirectly reflect the proliferation of cells. In T24 cells, the OD value after 5 days was significantly lower than that in si-NC group (Fig. 5A; P < 0.001). The same results were observed in 5637 cells (Fig. 5B; P < 0.001). In addition, the clone formation ability of T24 and 5637 cell lines was significantly inhibited in si-CDCA8 group compared with si-NC group (Figs. 5C–5G; P < 0.001). These results suggest that CDCA8 expression knockdown can inhibit the proliferation of bladder cancer cells.

Figure 5 Effect of knocking down CDCA8 on the proliferation of bladder cancer T24 and 5637 cells.

(A) Absorbance of OD450 in T24 cells was detected by CCK8 analysis; (B) Absorbance of OD450 in 5637 cells was detected by CCK8 analysis. (C–F) Clone formation experiment representative photos; (G) Quantitative statistical results. ***P < 0.001.

CDCA8 knockdown inhibits the migration and invasion of bladder cancer cells

We then investigated the effect of CDCA8 knockdown on the migration of T24 and 5637 cells through wound healing experiments. As shown in Figs. 6A–6J, the ability of cell migration in si-CDCA8 group was significantly inhibited compared with that in the si-NC group (P < 0.001).

Figure 6 Effect of CDCA8 knockdown on the migration and invasion of bladder cancer cells.

(A–E) Representational micrographs of wound healing experiments on T24 cells (100×) and migration statistics. (F–J) Representational micrographs of wound healing experiments on 5637 cells (100×) and migration statistics. (K–M) T24 representative micrographs of cell invasion experiments (100×) and its quantitative statistical results. (N–P) 5637 representative micrographs of cell invasion experiments (100×) and its quantitative statistical results. ***P < 0.001.

In addition, we conducted a Transwell assay to examine the effects of CDCA8 knockdown on the invasive ability of T24 and 5637 cells. The results showed that the number of invasive cells in the si-CDCA8 group was significantly lower than that in the si-NC group (Figs. 6K–6P; P < 0.001).

Effects of knocking down CDCA8 on the growth cycle of bladder cancer cells

To determine whether CDCA8 is involved in the regulation of T24 and 5637 cell growth cycles, we used flow cytometry to examine the effects of CDCA8 knockdown on the growth cycle of T24 and 5637 cells. The results showed that in T24 cells, the proportion of G0/G1 phase cells was decreased (P < 0.001), and the proportion of cells in S phase was increased (P < 0.01) after CDCA8 knockdown in si-CDCA8 group (Figs. 7A–7G). In 5637 cells, the proportion of G0/G1 phase cells in si-CDCA8 was decreased (P<0.001), whereas the proportion of cells in S phase and G2/M phase increased (P < 0.05), indicating that CDCA8 gene expression knockdown could result in 5637 cells S and G2/M phase growth arrest (Figs. 7H–7N).

Figure 7 Effect of CDCA8 knockdown on T24 and 5637 cell cycle.

(A–G) T24 cell cycle test results and statistical results; (H–N) 5637 cell cycle test results and statistical results. *P < 0.05, **P < 0.01, ***P < 0.001.

CDCA8 knockdown promotes the apoptosis of bladder cancer cells

After T24 and 5637 cells were successfully infected with lentivirus for 5 days, the apoptosis of T24 and 5637 cells was detected by flow cytometry. As shown in Fig. 8, the apoptosis rates of T24 and 5637 cells in si-CDCA8 group were significantly higher than those in si-NC group (P < 0.001), indicating that inhibiting CDCA8 expression could promote the apoptosis of bladder cancer cells.

Figure 8 Effect of knocking down CDCA8 gene on the apoptosis of bladder cancer cells.

(A–G) T24 cell apoptosis detection and statistical results; (H–N) 5637 cell apoptosis detection and statistical results. ***P < 0.001.

Analysis of the regulatory mechanism of CDCA8 in bladder cancer

In order to understand the regulatory mechanism of CDCA8 in bladder cancer, we calculated the CDCA8 co-expressed genes based on the transcriptome data of 414 bladder cancer samples in the TCGA database. The screening conditions were |Pearson correlation coefficient| > 0.4 and P < 0.01. A total of 978 genes were screened, including 171 negatively related genes and 807 positively related genes, and the top 20 positively related and top 20 negatively related genes were selected for expression heat map (Fig. 9A). We used the DAVID database (latest version 6.8, https://david.ncifcrf.gov) for GO functional annotation and KEGG pathway enrichment analysis. The GO function annotation includes three aspects: biological process (BP), molecular function (MF) and cell component (CC). The results showed that CDCA8 may have functions of regulating cell DNA binding, DNA repair and cell division in bladder cancer (Fig. 9B). KEGG pathway enrichment results showed that CDCA8 may affect the Cell cycle and P53 signalling pathways (Fig. 9C).

Figure 9 Analysis of the mechanism of CDCA8 in bladder cancer.

(A) Heat map of CDCA8 co-expressed genes; (B) GO function annotation of CDCA8 co-expressed genes; (C) KEGG pathway enrichment analysis of CDCA8 co-expression genes; (D) GSEA pathway analysis; (E) Analysis of the correlation between CDCA8 expression and RB1 mutation; (F) Analysis of the correlation between CDCA8 expression and RAS mutation; (G) Analysis of the correlation between CDCA8 expression and FGFR3 mutation; (H) Analysis of the correlation between CDCA8 expression and P53 mutation.

We divided the samples into two subgroups (high and low CDCA8 expression level subgroups) based on the CDCA8 gene expression values in these 414 bladder cancer samples and used GSEA analysis to predict the pathways that CDCA8 may affect. We used GSEA 3.0 software to calculate pathway enrichment scores and R language software to draw enrichment maps. We found several important pathways in the GSEA analysis results (Table 6; Fig. 9D), which are similar to those obtained using the co-expression analysis method.

Table 6 Analysis results of GESA in CDCA8 high expression group.

Pathway	ES	NES	NOM P-val	FDR q-val	
Cell cycle	0.787724	2.656093	0.000	0.000	
Bladder cancer	0.48828	1.841688	0.000	0.022	
Erbb signalling pathway	0.493038	1.957202	0.000	0.008	
P53 signalling pathway	0.544841	2.12953	0.000	0.001	
Pentose phosphate pathway	0.491114	1.901091	0.000	0.015	
Purine metabolism	0.534071	2.17924	0.000	0.001	
Pyrimidine metabolism	0.651945	2.24474	0.000	0.000	

In addition, we also used the GSE48075 dataset to analyse the correlation between CDCA8 expression and common mutant genes in bladder cancer. The results showed that CDCA8 overexpression was significantly associated with P53 mutation and RB1 mutation (Figs. 9E–9H).

Discussion

As one of the common malignant tumours in the urinary system, bladder cancer has become one of the most important health problems in the world. According to global cancer statistics, approximately 79,030 new cases of bladder cancer are diagnosed each year in the United States and 16,870 patients die of bladder cancer (Siegel, Miller & Jemal, 2018). Although bladder cancer can usually be cured by surgery and nonsurgical treatment, recurrence and metastasis remain the main challenges in clinical practice; bladder cancer recurrence remains as the most common cause of death in patients (Li et al., 2019). At present, the mechanism of occurrence and development of bladder cancer is not fully understood, and effective therapeutic targets and prognostic markers in the clinic are lacking (Gao et al., 2018). Various tumour biomarkers have been introduced in daily clinical practice, including risk assessment, screening, differential diagnosis, determination of prognosis, prediction of treatment response and monitoring of disease progression (Giunchi et al., 2017). Therefore, the identification of biomarkers related to the occurrence, development and prognosis of bladder cancer has important practical value for the clinical diagnosis and treatment of bladder cancer. Studying the role of protein-coding genes in bladder cancer from the molecular level is helpful to understand the mechanism of bladder cancer development and has important clinical relevance for the search for therapeutic targets and clinical prognosis markers in patients with bladder cancer. CDCA8, also known as Borealin/DasraB, forms a vertebrate chromosomal passenger complex (CPC) with Aurora B, INCENP and Survivin, which plays an important role in the localisation of CPC to centromere, stabilising the bipolar spindle and correcting the binding of kinetochore (Walker, 2001). To the best of our knowledge, the function of CDCA8 in bladder cancer cells has not been reported.

Gene chip, also known as DNA microarray, is a kind of biochip that has become an important means to obtain cancer gene expression profile information on a large scale. With the rapid development of science and technology, high-throughput sequencing technology has become a common method for the rapid search for molecular targets and prognostic markers for tumour drug therapy. An increasing number of tumour sequencing datasets promote the development of high-throughput sequencing databases, such as the GEO database (Barrett et al., 2013; Clough & Barrett, 2016).

Cell division cycle associated 8 is not only an oncogene to promote the occurrence and development of tumour but is also necessary for the proliferation and malignant development of cancer cells (Li et al., 2017; Yan et al., 2017). Jiao et al. (2015) also found that high CDCA8 expression can promote FOXM1 expression in breast cancer. The high expression of these two genes is significantly correlated with the poor prognosis of patients with breast cancer. They established a prognostic model of FOXM1-CDCA8 expression, which can be used to predict the prognosis of patients and may be a drug target for combined therapy of breast cancer. Hayama et al. (2007) also found that CDCA8 is co-activated and phosphorylated with AURKB in nonsmall cell lung cancer cells, and phosphorylated CDCA8 plays an important role in the growth and/or survival of cancer cells. The use of drugs to inhibit the CDCA8-AURKB pathway may be an effective treatment for patients with lung cancer. Ju et al. (2009) found that high CDCA8 expression is associated with the proliferation of epithelial ovarian cancer cells. Jeon et al. (2017) found that CDCA8 is highly expressed in HCC cells. The knockdown of CDCA8 expression by siRNA can inhibit the growth and colony formation of HCC cells by affecting cell cycle and promoting apoptosis. CDCA8 may serve as an effective target for the treatment of liver cancer in the future. In addition, CDCA8 gene is highly expressed in breast cancer (Phan et al., 2018), renal cell carcinoma (Gu et al., 2017), lung cancer (Hayama et al., 2007) and gastric cancer (Chang et al., 2006) and promotes the development of cancer, which is associated with the poor prognosis of these patients. All of these results indicate that CDCA8 is highly expressed in tumour and can promote tumour proliferation. Bi et al. (2018) analysed the CDCA8 expression in bladder cancer by using the GSE13507 dataset and found that high CDCA8 expression in bladder cancer is significantly related to the poor prognosis of patients; furthermore CDCA8 expression in bladder cancer was confirmed by qRT-PCR. This finding is consistent with our results. However, they did not elucidate the role of CDCA8 in bladder cancer cells. In the current study, we analysed the expression level of CDCA8 in bladder cancer by using four mRNA expression profile datasets in the GEO database, namely, GSE7476, GSE13507, GSE37815 and GSE65635. We also analysed the correlation between CDCA8 expression in bladder cancer and the prognosis based on the TCGA database. The analysis results showed that CDCA8 expression in bladder cancer was significantly higher than that of normal tissues. We also observed that the expression levels of CDCA8 were different in normal tissues in different datasets, which may be due to the inter-batch differences caused by different experimental methods and detection platforms for each dataset. However, this result did not affect the differential analysis of CDCA8 expression in a single dataset. We also analysed CDCA8 expression in different clinical factors. The results showed that CDCA8 expression was related to the progression of bladder cancer. The prognostic analysis showed that the high CDCA8 expression in the GSE13507 dataset was significantly related to the poor prognosis of patients. Multivariate COX regression analysis results showed that CDCA8 expression was independent of other clinical factors only in cancer-specific survival. This finding indicates that CDCA8 has good prognostic ability only when the tumour is the only lethal factor in patients. Notably, we also verified the expression of CDCA8 in bladder cancer and its clinical correlation by IHC. The IHC results also showed that CDCA8 expression in bladder cancer was significantly higher than that in normal tissues and was higher in patients with high stage and grade. This finding indicates that CDCA8 expression may be related to bladder cancer progression. These results suggest that CDCA8 is highly expressed in bladder cancer and may act as an oncogene to regulate the proliferation of bladder cancer cells and promote the development of bladder cancer. To further understand the role of CDCA8 in bladder cancer, we used T24 cells and 5637 cells to carry out the function experiment of CDCA8 in vitro. Knocking down CDCA8 could inhibit the proliferation, migration and invasion of bladder cancer cells. We also studied the effect of CDCA8 expression knockdown on cell cycle in T24 and 5637 cell lines, we found that knocking down CDCA8 expression can lead to S and G2/M phase arrest and inhibit the proliferation of bladder cancer cells. Notably, knocking down CDCA8 could induce the apoptosis of bladder cancer cells. In addition, we also used co-expression and GSEA methods to analyse the pathways that CDCA8 may affect in bladder cancer, and found that CDCA8 may participate in important pathways such as Cell cycle and P53 signalling pathway; We also found that the overexpression of CDCA8 was significantly related to the mutations of RB1 and P53. These results suggest that CDCA8 plays a key role in bladder cancer progression.

Conclusion

This study demonstrated that CDCA8 promotes the of bladder cancer via bioinformatics and in vitro cell experiments. CDCA8 knockdown can induce the apoptosis of bladder cancer cells. CDCA8 may serve as a potential therapeutic target for patients with bladder cancer. However, the specific regulatory mechanisms of CDCA8 in bladder cancer cells require further investigation.

Supplemental Information

Supplemental Information 1 Supplemental Files for Figure 3.

Click here for additional data file.

Supplemental Information 2 Supplemental Files for Figure 4.

Click here for additional data file.

Additional Information and Declarations

Competing Interests

Author Contributions

Human Ethics

Data Availability

The authors declare that they have no competing interests.

Xin Gao conceived and designed the experiments, performed the experiments, analysed the data, prepared figures and/or tables, authored or reviewed drafts of the paper, and approved the final draft.

Xiaohong Wen conceived and designed the experiments, performed the experiments, analysed the data, prepared figures and/or tables, authored or reviewed drafts of the paper, and approved the final draft.

Haowei He performed the experiments, prepared figures and/or tables, and approved the final draft.

Linlin Zheng performed the experiments, prepared figures and/or tables, and approved the final draft.

Yibo Yang performed the experiments, prepared figures and/or tables, and approved the final draft.

Jinlian Yang performed the experiments, prepared figures and/or tables, and approved the final draft.

Haifang Liu performed the experiments, authored or reviewed drafts of the paper, and approved the final draft.

Xiguo Zhou performed the experiments, prepared figures and/or tables, and approved the final draft.

Changshun Yang performed the experiments, authored or reviewed drafts of the paper, and approved the final draft.

Yinyi Chen analysed the data, prepared figures and/or tables, and approved the final draft.

Mei Chen performed the experiments, authored or reviewed drafts of the paper, and approved the final draft.

Shufang Zhang conceived and designed the experiments, performed the experiments, authored or reviewed drafts of the paper, and approved the final draft.

The following information was supplied relating to ethical approvals (i.e., approving body and any reference numbers):

All sample collections were approved by the Haikou Municipal People’s Hospital Ethics Committee under approval number SC20180238.

The following information was supplied regarding data availability:

The raw data is available at Figshare: gao, xin (2020): Rawdata.zip. figshare. Dataset. DOI 10.6084/m9.figshare.10311638.v1.

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
