# Peer review of "Knockdown of CDCA8 inhibits the proliferation and enhances the apoptosis of bladder cancer cells"

_PeerJ, doi:10.7717/peerj.9078_

## Round 0.1 · original submission · Major Revisions

The article is interesting but major revision is needed. As suggested by reviewers, additional in vitro experiments to confirm the findings, extension of analysis to other clinical dataset (like TCGA), addition of multivariate analysis including clinical varibailities, improvements in discussion and language editing would help to improve the quality of the manuscript.

Reviewer 1 ·

Basic reporting

No comment.

Experimental design

No comment.

Validity of the findings

No comment.

Additional comments

The present study has found an oncogene called CDCA8 (cell division cycle associated 8) which could promote proliferation and inhibit apoptosis in bladder cancer. They used bioinformatics analysis as a premise, combing with the in vitro experiment to complete the study. The results are convincing and make some sense, but I still have some suggestions to improve the article:
1. The length of the introduction is too short. Please search more reference to support your idea.
2. The phrase “RT-qPCR assay” is wrong.
3. In spite of GSE13507, more datasets such as TCGA-BLCA, GSE32584 and GSE48075 etc. should be analyzed to see whether CDCA8 is upregulated and correlated with poor prognosis in all this datasets.
4. The overexpression experiment should be added to validate the role of CDCA8 in bladder cancer.
5. The western blot of marker proteins such as CCND1, caspase etc. should be done.

·

Basic reporting

1. The sentence “The expression of … was knocked down” is not accurate (line 41, for example). It is rather the gene that is knocked down and not its expression
2. Sentences at lines 67, 82 have no reference.
3. Line 77. When referring to the authors of an article by name “Dai et al.” the intext citation that follows should be only the year in parentheses to avoid repetition, unless the journal has explicit format that cover this case. Alternatively, the full intext citation can be moved to the end of the sentence.
4. Line 89. “... analyzed by bioinformatics.” The authors should consider being more specific about with methods and tools were used to analyze the correlation.
5. The GEO datasets were wrongly referred to in the text as “high throughput sequencing datasets ” (line 87) and “chip sequencing data” (line 94). The four datasets were generated using microarrays!
6. The method to obtain the datasets and the form of the data (raw/processed) were not mentioned.
7. The transformation of gene probe ID to gene symbol (line 99) is rarely as simple as that. How did the authors map probes from different platforms to gene symbols? How did the authors deal with probes mapping to multiple genes and genes represented by multiple probes? Also, “international standard gene names” doesn’t refer to any specific naming convention/standard.
8. Line 119. The term WB was not mentioned in full.
9. The calculation of the delta delta Ct value was described as subtracting the mean delta Ct of control from each sample separately. Some argue that the means of two groups should be used. Please justify or cite the source. Also the calculation of the error term was not described.
10. The term “dataset” and “data set” were both used in the text. Similarly, the terms “knockdown” and “knockout” were used interchangeably. Please revise for consistency.
11. Figure 1. The x-axis doesn't start from 0 and the figure legend is very brief.
12. Figure 3, 6-8 legend are very brief.

Experimental design

1. The authors chose four independent datasets of normal and bladder cancer tissue to visualize the expression of CDCA8 and associate it with clinical characteristics and outcomes. The choice of these particular datasets was not justified. Large studies such as those of the cancer genome atlas can be a better choice if the main goal is to establish links between the expression and patient metadata.
2. The correlation of gene expression with clinical characteristics was not formally analyzed. Although figure 2 provide useful visualization of the expression in different groups, to claim significant correlations, they should be tested in regression models.
3. Gene and protein markers were not used to study the effect of the knockdown of CDCA8 on cell cycle or growth.

Validity of the findings

1. The visualization of the expression of CDCA8 in multiple bladder cancer microarrays datasets is valid. However, the claim that this gene correlates with the tumor, stage, grade and age was not well established. The use of multiple regression model is required to establish the associations.
2. The functional study of the CDCA8 knockdown bladder cancer cell lines is suggestive. However, the mechanism by with this protein might be working was not fully investigated. The cell cycle analysis is useful but doesn’t justify the strong claims. First, analysis of key gene and protein markers of proliferation, migration, invasion, apoptosis and cell cycle is missing. Second, pattern of cell cycle regulation is in one cell line is opposite to the other. At least, this difference should be addressed in the discussion.

Reviewer 3 ·

Basic reporting

.

Experimental design

.

Validity of the findings

.

Additional comments

Authors of the present study investigated the correlation between the expression of
CDCA8 in bladder cancer and clinical variables as well as the role of CDCA8 in bladder cancer cells in vitro. The aim of the study is original. However, some revisions are needed in order to improve the overall quality of the manuscript.
Methodology: I suggest to improve the methodology by analyzing, in a multivariable model including also clinical variables, the predictors of survival.
Discussion: I suggest to improve the discussion about the significant correlations found between CDCA8 expression and some clinical variables.

Reviewer 4 ·

Basic reporting

The article is written in a clear and concise manner. The introduction has depth with good background information. The authors have, however, skipped referencing a paper - Bi et al, 2018 “CDCA8 expression and its clinical relevance in patients with bladder cancer” which is very closely related to this article. It would be good for them to reference this article and potentially in the discussion compare and contrast their results with this article. The discussion is a little repetitive and may be overstating the use of CDCA8 as a prognostic marker as high expression is seen in many patients with good prognosis. Good detail in the materials and methods regarding the experimental techniques; but more detail is necessary regarding computational methods/bioinformatics with respect to the different analysis performed. It would be worthwhile to compress the figures from 9 to 4 to allow for a more streamlined flow of the paper and the data. For instance, the migration and transwell assays can easily be included as the same figure and similarly for the comparison of levels of CDCA8 and clincal features and survival curves. More detail can be added to the figure legends of figure 5,6,7,8,9. Some editing for language and grammatical errors would be useful before the publishing of this manuscript.

Experimental design

The experimental design is suitable to the goal that the authors have outlined and the article is within the scope of the journal. The authors have described a knowledge gap in the field by stating that role of CDCA8 in bladder cancer in unknown. By stating the role of CDCA8 in tumorigenesis in other cancers, the authors justify their preliminary notion that CDCA8 might have an important role to play in bladder cancer as well. Using bioinformatic analysis of published microarray data as well with comparison with clinical data of those samples, the authors have identified that CDCA8 is overexpressed in bladder cancer cells relative to normal cells. Further, they used in vitro assays and identified that CDCA8 knockdown in the bladder cancer cells results in the reduced proliferation, migration, invasion, cell cycle arrest and increased apoptosis. While the authors have performed these in vitro studies using two different cell lines, no information is provided regarding the number of biological replicates done and whether the quantification is representation of multiple replicates. This would be useful to include in the final version of the article.

Validity of the findings

The authors aim to identify the role of CDCA8 in bladder cancer. By using a two pronged approach of bioinformatics and in vitro assays, they provide good evidence to support their hypothesis. The following changes/experiments would be useful to strengthen the paper.
1) In Figure 1, the authors identify that CDCA8 expression is increased in bladder cancer tissue relative to normal tissue. It would be useful to provide more information regarding the studies that were used. It can also be noted that the levels of CDCA8 in normal tissues varies amongst the studies as well as the number of normal tissues used for comparison are very low – a sentence or two of comment regarding this would be useful. Finally, given the availability of large resources now such the TCGA as well as other large RNAseq dataset, it would be good to see corroborative evidence from the analysis of a larger dataset or justification as to why this was not done. Lastly, comparison of levels of CDCA8 to known markers of bladder cancer as well negative controls (house keeping genes) would provide strength to the study showing the CDCA8 is indeed a marker for bladder cancer.
2) In Figure 2, the authors use both p=value and sometimes p<value in different sub-figures; this would need to changed to be of the same format in all the figures with a clear NS written in graphs describing non-significant comparisons. Further, the authors claim that the expression of CDCA8 is highly and significantly correlated with a clinical factor – however, no correlation analysis has been performed to justify this claim. I would recommend that the language be amended to “ CDCA8 expression is higher in individuals with high grade tumor when compared to individuals with low grade tumor” etc.
3) In figure 4, an explanation needs to provide in the methods or results regarding what is driving the GFP expression and how the the CDCA8 knockdown results in a loss of GFP expression. While the mRNA expression and protein expression are convincing regarding the CDCA8 knockdown, the knockdown is not apparent looking at the fluorescence images.
4) In figure 5, while the effects on CDCA8 knockdown on the growth of these cells is striking, corroborative experiments such as Brdu incorporation etc would be useful to add strength to the claim that these cells have reduced proliferation.

---

## Round 0.2 · Minor Revisions

Authors have addressed the majority of comments but reviewers have recommended minor revisions for some minor issues to answer or discuss. We kindly ask you to answer these questions to improve the quality of manuscript.

Reviewer 1 ·

Basic reporting

The author have improved their writing.

Experimental design

No comment.

Validity of the findings

No comment

Additional comments

1. I think the dataset GSE48075 do include patient prognosis information as I have used this dataset in my study. So I also recommend author to make some analysis in GSE48075.
2. I think some markers associated with cell cycle, apoptosis should be done to prove CDCA8 really influence these process.

·

Basic reporting

no comment

Experimental design

no comment

Validity of the findings

no comment

Additional comments

The revised manuscript was improved as a result of:
Using TCGA data to visualize the expression of CDCA8
Performing univariate and multivariate analysis to establish CDCA8 as prognostic factor

The authors chose to not extend the study into the mechanism by which this protein might influence the proliferation or the cell cycle of bladder cancer. While this represents an issue of the rigour of the work, it has been acknowledged in the discussion and the conclusion.

Two points remains to be resolved however. First, no correlation was observed between CDCA8 expression and the patient prognosis. I’d be inclined to further investigate this point. Second, two cell lines had opposing patterns of cell cycle regulation. While this was reported as such, no explanation was provided besides “may be CDCA8 has different effects on the cell cycle.” This might be true, but it basically contradicts the rest of the study. Therefore, observing the desired pattern in the other cell line cannot be taken as supporting evidence.

---

## Round 0.3 · accepted · Accept

Your manuscript has been improved with last revision and it is now acceptable for publication.